# The Potential of Neoagaro-Oligosaccharides as a Treatment of Type II Diabetes in Mice

**DOI:** 10.3390/md17100541

**Published:** 2019-09-20

**Authors:** Fudi Lin, Dongda Yang, Yayan Huang, Yan Zhao, Jing Ye, Meitian Xiao

**Affiliations:** 1College of Chemical Engineering, Huaqiao University, Xiamen 361021, China; 15105966539@163.com (F.L.); 17359896913@163.com (D.Y.); yyhuang@hqu.edu.cn (Y.H.); 18404985982@163.com (Y.Z.); 2Xiamen Engineering and Technological Research Center for Comprehensive Utilization of Marine Biological Resources, Xiamen 361021, China

**Keywords:** type 2 diabetes mellitus, neoagaro-oligosaccharides, MAPK, Nrf2, PPARγ

## Abstract

Type 2 diabetes mellitus (T2DM) accounts for more than 90% of cases of diabetes mellitus, which is harmful to human health. Herein, neoagaro-oligosaccharides (NAOs) were prepared and their potential as a treatment of T2DM was evaluated in KunMing (KM) mice. Specifically, a T2DM mice model was established by the combination of a high-fat diet (HFD) and alloxan injection. Consequently, the mice were given different doses of NAOs (100, 200, or 400 mg/kg) and the differences among groups of mice were recorded. As a result of the NAOs treatment, the fasting blood glucose (FBG) was lowered and the glucose tolerance was improved as compared with the model group. As indicated by the immunohistochemistry assay, the NAOs treatment was able to ameliorate hepatic macrovesicular steatosis and hepatocyte swelling, while it also recovered the number of pancreatic β-cells. Additionally, NAOs administration benefited the antioxidative capacity in mice as evidenced by the upregulation of both glutathione peroxidase and superoxide dismutase activity and the significant reduction of the malondialdehyde concentration. Furthermore, NAOs, as presented by Western blotting, increased the expression of p-ERK1/2, p-JNK, NQO1, HO-1, and PPARγ, via the MAPK, Nrf2, and PPARγ signaling pathways, respectively. In conclusion, NAOs can be used to treat some complications caused by T2DM, and are beneficial in controlling the level of blood glucose and ameliorating the damage of the liver and pancreatic islands.

## 1. Introduction

Type 2 diabetes mellitus (T2DM) accounts for more than 90% of cases of diabetes mellitus. It is a polygenic disease characterized by insufficient insulin secretion caused by the destruction of islet β-cells and insulin resistance. A complicated syndrome is common among the individuals suffering from T2DM. Oxidative stress can be found in the patients during the development of T2DM. Particularly, the lack of balance between the levels of free radicals and the antioxidant defense system always leads to damage of tissues and β-cells [1,2,3,4,5]. Among the tissues, the liver plays a central role in lipid and glucose metabolism. Its dysregulation can result in various diseases, including diabetes [6]. Moreover, T2DM is accompanied by comprehensive disorders in the metabolism of glucose, protein, and lipid [7,8]. As one of the typical metabolism disorders, dyslipidemia is prevalent among T2DM patients, involving not only high levels of serum triglycerides (TG), total cholesterol (TC), and low-density lipoprotein cholesterol (LDL-C), but also a low concentration of high-density lipoprotein cholesterol (HDL-C). In addition, cardiovascular complications should be taken into account when treating T2DM [4,9,10,11].

As the hydrolysate of agar/agarose, neoagaro-oligosaccharides (NAOs) are prepared by breaking the β-(1–4) bonds using β-agarase; they display various bioactivities including antioxidant [12,13,14,15], anti-inflammatory [16,17,18], whitening of melanoma cells [19,20], anti-obesity [21], and hypolipidemic effects [22]. Two NAOs, neoagaroteraose (signed as NA4) and neoagarohexaose (signed as NA6), have been reported to be able to ameliorate obesity and the obesity-related metabolic defects, such as hyperlipidemia, steatosis, insulin resistance, and glucose intolerance in obese mice, by means of the induction of adiponectin [21]. Moreover, NA4 was more effective in scavenging hydroxyl radicals when compared with neoagarobiose (signed as NA2), NA6, and neoagarooctose (signed as NA8). NA4 also inhibited the inflammation in lipopolysaccharide (LPS)-stimulated macrophages by suppressing the MAPK and nuclear factor-κB (NF-κB) pathways [23]. Moreover, NAOs could stimulate the expression of A20 and COX-2 via MAPK and NF-κB-dependent pathways in septic shock [24]. It is interesting that the defects mentioned above that might be treated by NAOs are also common among T2DM patients. Whether NAOs play a passive role in the treatment of T2DM deserves attention.

Although the effects of NAOs on obesity and obesity-related metabolic defects had been described, their efficacy in the treatment of diabetes mellitus and the involved molecular mechanism were still unknown. Therefore, in this study, the potential of NAOs in treating T2DM was evaluated in KM mice, while the corresponding anti-T2DM mechanism of NAOs was also investigated.

## 2. Results

### 2.1. Characterization of the NAOs

Agar was hydrolyzed by β-agarase and the hydrolyzate was purified by means of centrifugation, membrane separation, and gel column separation, generating the target NAOs with acceptable purity. The composition of the purified oligosaccharides was determined by using HPLC-ELSD. As shown in Figure 1A, NA2 and NA4, the structures of which are also presented in the same figure, were identified. The NAOs mixture, composed of NA2 (21.6%) and NA4 (71.1%), could be well separated in the HPLC experiment. Besides, the structures of the NAOs were confirmed by using ESI-TOF-MS and ^13^C-NMR. The ESI-TOF-MS results, as presented in Figure 1B, exhibited the expected mass of NA2 (*m*/*z*: calculated, 324.1056 for C_12_H_20_O_10_; found, 347.0949 for [M + Na]^+^) and NA4 (*m*/*z*: calculated, 630.2007 for C_24_H_38_O_19_; found, 653.1886 for [M + Na]^+^). The ^13^C-NMR results (Figure 1C) showed the distinctive chemical shifts as 92.6 ppm and 96.6 ppm, of which were the signals of the α- and β-anomeric forms of the galactose unit at the reducing end of the NAOs, respectively. However, sufficient NA2 and NA4 were unable to be obtained by using preparative liquid chromatography due to the high cost. Therefore, the mixture was used to perform the remaining experiments.

### 2.2. Effects of NAOs on Body Mass and Food Intake

Body mass and food intakes were recorded to explore the influence of NAOs on growth. During this experiment, the body weight of the mice in each group increased gradually (Figure 2). The body mass of the model control (MC) group was 29.2 ± 1.3 g, much lower than that of the normal control (NC) group (*p* < 0.05), suggesting the typical body weight loss in T2DM. After eight weeks of treatment, administration of NAOs (100, 200, or 400 mg/kg) or that of metformin (50 mg/kg) was able to recover the body weight of mice as compared with the MC group (*p* < 0.01). Nevertheless, little difference of the food intake could be found in each group as shown in Table 1. These results indicate that NAOs might prevent the body weight loss caused by T2DM without any change of food intake.

### 2.3. Fasting Blood Glucose (FBG) Determination and Oral Glucose Tolerance Test (OGTT)

The concentration of FBG was determined to study the effects of NAOs on the glucose status of T2DM mice. As shown in Table 2, the level of FBG in the MC group was much higher than that in the NC group (22.24 ± 5.76 vs. 4.83 ± 0.69, *p* < 0.01) at the end of the study, suggesting some diabetes-induced defects. Metformin (50 mg/kg) generated powerful hypoglycemic effects on mice, while NAOs could reduce the glycemia in the diabetic mice in a dose-dependent manner. The FBG levels in the M-NAO and H-NAO groups were significantly reduced by 29.0% (*p* < 0.05, vs. MC group) and by 41.8% (*p* < 0.01, vs. MC group), respectively. NAOs were able to protect mice from the hyperglycemia caused by T2DM.

OGTT was performed to investigate the effect of NAOs on glucose tolerance. As presented in Figure 3A, the blood glucose concentration in the NC group rose to its peak within 30 min and it returned to its initial level within the next 90 min. A much higher glucose level was found in the MC group, with its peak value of 31.2 ± 3.2 mmol/L within the first 30 min, while the glucose level was still high at the end of the OGTT. However, the rise of the blood glucose caused by the glucose was inhibited by the NAOs treatment in a dose-dependent manner. The area under the curve (AUC) of the M-NAO and H-NAO groups in Figure 3A was markedly lower than that of the MC group (*p* < 0.01 and *p* < 0.001, respectively) (Figure 3B). NAOs was able to improve the glucose intolerance of the diabetic mice.

### 2.4. Immunohistochemistry of the Liver and Pancreatic Islets

Immunohistochemical staining revealed that the liver of mice fed with a normal diet was normal while hepatic steatosis developed in mice that were HFD-fed (Figure 4A,B). However, NAOs treatment ameliorated the macrovesicular steatosis and hepatocyte swelling induced by HFD in a dose-dependent manner (Figure 4D–F). These results suggest that NAOs can inhibit fat deposition in the liver.

The histology of the pancreas was also studied. As shown in Figure 5A,B, the injection of alloxan caused pancreatic islet pathology in mice as evidenced by severe narrowness of this organ. However, metformin (as shown in Figure 5C) and different doses of NAOs (as show in Figure 5D–F) were capable of weakening the reduction. In addition, the number of β-cells in the pancreas was studied. As shown in Figure 6, the β-cell number in the MC group (45 ± 13) was much less than that of the NC group (214 ± 22) due to the destruction of the pancreas caused by alloxan. However, the reduction of β-cells might be substantially ameliorated by either metformin or NAOs. Specifically, the number of islet cells in the L-NAO, M-NAO, and H-NAO groups were 89 ± 16, 116 ± 23, and 126 ± 27, respectively, higher than that in the MC group. There is no doubt the NAOs treatment can retard alloxan-induced islet cell destruction in mice.

### 2.5. Effects of NAOs on Serum Lipids

The level of serum lipids was determined in order to evaluate the anti-hyperlipidemic potential of NAOs. As presented in Table 3, a HFD resulted in higher concentrations of TG, TC, and LDL-C and lower levels of HDL-C when compared with a normal diet (*p* < 0.01). Nevertheless, NAOs at medium dose or high dose might reduce the concentrations of TG, TC, and LDL-C. Interestingly, the level of HDL-C might also be recovered by NAOs in the H-NAO group. Taken together, NAOs exhibited potent anti-hyperlipidemic activity.

### 2.6. Effects of NAOs on the Hepatic Antioxidation System

Oxidative stress plays an essential role in the development of T2DM, and antioxidant enzymes are the major bioactive substances dealing with reactive oxygen species (ROS) [25]. The levels of MDA, GSH-Px, and SOD were investigated to explore the antioxidant activities of NAOs for a deeper understanding of the anti-diabetic effects of NAOs. More MDA was found in the MC group as compared with NC group, indicating the ROS damage. However, the level of MDA could be reduced by both metformin (50 mg/kg) and NAOs (200 mg/kg or 400 mg/kg). More importantly, GSH-Px and SOD, two important antioxidant enzymes, were up-regulated by NAOs at the dose no less than 200 mg/kg (*p* < 0.01 vs. MC). It is obvious NAOs are a benefit to the antioxidant system in T2DM mice.

### 2.7. NAOs Regulated the MAPK-Nrf2 Pathway

The MAPK signaling pathway plays an important role in the development of T2DM. Therefore, some key proteins involved in the pathway were investigated in the presented study. First of all, the effects of NAOs treatment on the phosphorylation levels of ERK1/2, JNK, and p38 MAPK were determined. A HFD resulted in higher phosphorylation of ERK1/2 and JNK compared with the NC group. However, eight weeks of NAOs treatment significantly ameliorated the abnormal phosphorylation of ERK1/2 and JNK (Figure 7A,C) with little change of p-p38MAPK (Figure 7A,D). Moreover, the expression of nuclear factor erythroid-2-related factor 2 (Nrf2), oxygenase 1 (HO-1), and NAD(P)H quinone dehydrogenase 1 (NQO1) were explored. MAPK signaling also affects Nrf2, always resulting in the transcription of the genes encoding HO-1 and NQO1. As shown in Figure 8, as expected, the levels of hepatic HO-1 and NQO1 in the NAOs-treated groups were markedly higher than those in the MC group. The results imly that NAOs might reverse hepatic injury in T2DM by regulating the expression of some key hepatoprotective genes.

### 2.8. NAOs Regulated the PPARγ Pathway

For better comprehension of the hepatoprotective mechanism of NAOs in diabetic mice, PPARγ expression was also studied. The results presented in Figure 9 show that the PPARγ expression was potently suppressed when KM mice were treated with a HFD and alloxan in the MC group, while the suppression was weakened by using 50 mg/mL of metformin. The NAOs treatment, however, was also able to raise the expression of PPARγ. These results imply that NAOs prevent T2DM mice from hepatic damage partly by promoting the expression of PPARγ.

## 3. Discussion

NAOs are the hydrolysates of agar/agarose, and have been reported to have beneficial effects to animals subjected to metabolic diseases [21,22]. Herein, a NAOs mixture composed of NA2 (21.6%) and NA4 (71.1%) was prepared and its potential in treating T2DM was investigated in mice. The study indicated that NAOs can promote the growth of the mice suffering from T2DM. The mixture of NAOs also possessed potent abilities of both reducing FBG and improving glucose tolerance in a dose-dependent manner. Further investigation showed the capacities of NAOs were closely connected to their ability to recover the insulin level in T2DM mice. This was supported by the finding that NAOs was able to prevent the pancreatic tissue from the shrinking caused by T2DM and they could recover the numbers of islet cells. Apparently, NAOs might exhibit a positive influence on the glucose metabolism system of T2DM mice.

Dyslipidemia is one of the metabolic syndromes often occurring with T2DM. It can lead to severe circulatory system dysfunction. High levels of TG, TC, and LDL-C and a low level of HDL-C contribute to the development of coronary artery disease in the patients with uncontrolled T2DM [26,27,28,29,30]. In the present study, the high levels of TG, TC, and LDL-C induced in T2DM could be reduced by NAOs, while the mixture might also raise the concentration of HDL-C. Obviously, NAOs might be used to treat the dyslipidemia that accompanies dysglycemia, which would be very good for the treatment of T2DM.

The lack of balance between free radicals and antioxidant enzymes plays an essential role in the development of T2DM. Free radicals generated under oxidative stress can cause defects in glucose metabolism. The radicals have a detrimental impact on pancreatic β-cell function, leading to hyperglycemia. In the presented study, MDA, one of the major lipid peroxides induced by radicals, was greatly reduced by NAOs in the T2DM mice, implying the antioxidant effect of NAOs. Furthermore, the activity of antioxidant enzymes can be inhibited in T2DM, leading to cell and tissue damage [31,32,33]. However, the abilities of the enzymes were up-regulated by NAOs in this study. SOD, an important antioxidant enzyme metabolizing the superoxide anion to form hydrogen peroxide, was found to be reduced in T2DM [25,34], which hinders the clearance of free radicals. Likely, the reduction of SOD can accelerate the progression of T2DM. However, the level of SOD was significantly increased by NAOs in this study (Table 4). Another key antioxidative enzyme, specifically GSH-Px, able to scavenge free radicals by means of transforming reduced GSH into its oxidized form [34], was investigated as well. Although little difference of GSH-Px activity was observed between T2DM patients and the normal control, the up-regulation of GSH-Px could be carried out in the T2DM mice by giving mice NAOs (Table 4). This suggests NAOs can retard the liver damage of T2DM mice caused by oxidative stress.

MAPK pathways are extremely important to the metabolic effects of insulin because they can convent extracellular signals into specific cellular responses through a series of phosphorylation events. Among the proteins involved in the MAPK pathway, ERK1/2, JNK, and p38 MAPK play a crucial role in activating the MAPK signaling cascades [35], the phosphorylation of which was promoted by alloxan in this study, and are involved in hepatic injury. However, NAOs treatment was able to down-regulate the phosphorylation of ERK and JNK in the presence of a HFD and alloxan. In addition, the effect of NAOs on Nrf2 was investigated due to the fact that the expression of Nrf2 was controlled by the MAPK pathways including ERK, JNK, and p38 [36,37,38]. In the presented study, the inhibition of either ERK or JNK abolished NAOs-induced up-regulation and phosphorylation of Nrf2, demonstrating the two pathways might mediate Nrf2 activation in NAOs-treated mice. Furthermore, NQO1 and HO-1, the expression of which can be increased by the release of Nrf2 and its entry into the nucleus [38,39], were also explored. The levels of these two enzymes might be recovered by NAOs treatment, indicating the activity of NAOs in alleviating oxidative stress in vivo. This suggests that after the treatment with NAOs, ERK/JNK and NQO1/HO-1 are primarily activated and a positive-feedback loop with each other is then formed to protect the mouse’s liver from the oxidative damage induced by high blood glucose.

PPARγ is recognized as a master regulator of adipogenesis, which involved in the normal function of fat cells [39]. The severe hyperglycemia presented in the individuals undergoing dominant-negative mutations in the PPARγ gene have already confirmed a genetic link between PPARγ and T2DM [40]. As compared with the obese subjects with T2DM, the expression of PPARγ is higher in the non-diabetic obese patients [41]. Herein, the mice fed a HFD were found to suffer from fatty livers, with weak expression of PPARγ. However, significantly higher expression of hepatic PPARγ was observed in the NAOs treatment groups as compared with the model group. This suggests that NAOs may ameliorate the abnormal lipid metabolism through modulation of PPARγ expression, which might keep the mice away from fatty liver disease. It was evident that NAOs might hinder the lipid accumulation in obese mice by regulating hepatic lipid metabolism.

## 4. Materials and Methods

### 4.1. Materials and Animals

The assay kits for measuring total protein, TG, TC, HDL-C, LDL-C, MDA, Insulin, GSH-Px, and SOD were purchased from Jiancheng Bio-engineering Institute (Nanjing, China). The primary antibodies against ERK1/2, phospho-ERK1/2 (p-ERK1/2), JNK, phospho-JNK (p-JNK), p38 MAPK, phospho-p38MAPK (p-p38 MAPK), Nrf2, NQO1, HO-1, PPAR-γ, and glyceraldehyde-3-phosphate dehydrogenase (GAPDH) were purchased from the Proteintech Group (Chicago, IL, USA), and horseradish peroxidase (HRP)-linked anti-rabbit IgG secondary antibodies were obtained from Cell Signaling Technologies (Danvers, MA, USA). Polyvinylidene fluoride (PVDF) membranes were from Millipore (Billerica, MA, USA). Biochemical grade reagents for Western blotting (WB) were purchased from Beyotime Biotechnology (Shanghai, China). The agar used, composed of agarose and agaropectin, was obtained from Fujian Jinyan Marine Biotechnology Co., Ltd. (Fujian, China), with had a purity of more than 90%. Other reagents of analytical grade were purchased from commercial suppliers. Four-week-old specific pathogen-free male KM mice of 18–22 g body mass were purchased from Shanghai SLAC Laboratory Animal Co., Ltd., with license number SCXK (Shanghai, China) 2012-0002. Mice were housed in a standard sterile animal room with controlled temperature (25 °C) and humidity (50%–60%) and a 12 h light/dark cycle. Permission was obtained to conduct the animal experiments described (approval number and date: 2017/HQU and 06 March 2017) and all the experiments followed the guidelines of the Ethics Committee of Huaqiao University.

### 4.2. Preparation of NAOs

Agar was added to Tris-HCl buffer at 0.3% *w*/*v* and the mixture was heated for 10 min in a boiling water bath. Crude β-agarase was added to the cooled agar solution [42,43], and the mixture was incubated in an incubator with shaking (120 rpm; Hunan Xiangyi Laboratory Instrument Development Co., Ltd., H1850, Hunan, China) at 45 °C for 24 h, after which the insoluble component was removed by centrifugation (Shanghai Xinrui Automation Equipment Co., Ltd., Shanghai, China) at 12,000 × *g* for 30 min. Membrane separation included ultrafiltration to remove proteins and nanofiltration to remove small molecules such as salts, aiming to separate and purify the NAOs, which were further purified by gel column. The solution was concentrated and freeze-dried, and the NAOs were analyzed using an HPLC-ELSD system (Waters, Milford, MA, USA). ESI-TOF-MS and NMR were utilized to explore the structure and molecular mass of the NAOs, respectively [43].

### 4.3. Experimental Design

All mice were acclimated to their surroundings for 1 week, and then some mice were fed a normal diet while others were fed a HFD for 2 weeks. Then, T2DM was induced by alloxan in HFD-mice. All mice were divided into six groups (*n* = 10/group) as follows: a normal control (NC) group without diabetes; a model control group (MC) with diabetes but no treatment; a positive control group (PC) with diabetes and treatment with 50 mg/kg metformin; a low-dose NAOs group (L-NAO) with diabetes and treatment with 100 mg/kg NAOs; a medium-dose NAOs group (M-NAO) with diabetes and treatment with 200 mg/kg NAOs; a high-dose NAOs group (H-NAO) with diabetes and treatment with 400 mg/kg NAOs by gavage. Then, the MC, PC, L-NAO, M-NAO, and H-NAO groups were fed a HFD (Table 5), while the NC group continued to consume normal chow for 8 weeks. Oral administration of each substance was performed daily for 8 weeks. The mice were allowed free access to food and water throughout the experiment.

The body mass and food intake of the mice were recorded during the experiment. Tissue masses were measured and organ indexes were calculated and defined as the ratios of each tissue/organ mass to body mass. Portions of liver and pancreas were fixed in 10% neutral-buffered formalin (NBF) for subsequent sectioning and histological assessment, and the remaining tissues were frozen in liquid nitrogen and stored at −80 °C until analysis.

### 4.4. FBG and OGTT

Every 4 weeks, mice were fasted for 12 h and the glucose concentration in blood was obtained from the tail vein. After 8 weeks of NAOs administration, FBG was measured and OGTT was performed as described previously [21]. Briefly, the mice were fasted for 12 h, FBG was determined, and then an oral dose of 2 g/kg glucose was administered. The concentrations of blood glucose at 0, 30, 60, 90, and 120 min were then measured using a blood glucose meter (Accu-check Active, Roche, Berlin, Germany). A glucose concentration-time plot was then prepared to compare the changes in glucose with time for each treatment and to calculate the integrated area under the curve (AUC) during the OGTT, using the following equation:AUC = (BG_0 min_ + 2 × BG_30 min_ + 2 × BG_60 min_ + 2 × BG_90 min_ + BG_120 min_) × 0.5 h × 0.5(1)
where BG_0 min_, BG_30 min_, BG_60 min_, BG_90 min_, and BG_120 min_ are the blood glucose concentrations measured at 0, 30, 60, 90, and 120 min, respectively.

### 4.5. Biochemical Analysis

After 8 weeks of treatment with NAOs, all the mice were fasted for 12 h and blood was collected into heparinized tubes. Serum samples were obtained by centrifugation at 3500 × *g* for 15 min at 4 °C. The serum concentrations of TG, TC, HDL-C, LDL-C, and insulin were then enzymatically determined using Nanjing Jiancheng Bioengineering Institute (Nanjing, China).

### 4.6. Determination of GSH-Px, SOD, and MDA

Liver samples were rinsed and homogenized (1:10, *w*/*v*) in cold 50 mM Tris buffer (pH 7.4). The homogenates were then centrifuged at 12,000 × *g* for 5 min at 4 °C, and GSH-Px and SOD activity and MDA concentration were measured in the supernatant, following the instructions provided in Nanjing Jiancheng Bioengineering Institute (Nanjing, China).

### 4.7. Histologic Assessment of the Liver and Pancreas

Liver and pancreas samples were washed with cold normal saline, fixed in 10% NBF, and embedded in paraffin. These paraffin blocks were sectioned, stained with hematoxylin and eosin (H&E), and the histologic appearance was assessed using an optical microscope (Leica Microsystems, 090-135.002, Heidelberg, Germany).

### 4.8. Western Blot Analysis

Liver samples were homogenized in a cell lysis buffer containing a protease inhibitor cocktail and phenylmethylsulfonyl fluoride. The protein concentration of each lysate was measured using a protein assay kit in accordance with the manufacturer’s protocol, and then equalized by addition of the appropriate volume of loading buffer, followed by denaturation by boiling for 5 min. To determine the relative protein expression of ERK1/2, p-ERK1/2, JNK, p-JNK, p38 MAPK, p-p38 MAPK, Nrf2, NQO1, HO-1, and PPAR-γ, samples containing 25 μg of protein were separated by 10% sodium dodecyl sulfate-polyacrylamide gel electrophoresis and then transferred to PVDF membranes. The membranes were blocked with 5% non-fat milk in Tris-buffered saline containing 0.1% Tween-20 (TBST) for 1 h, then washed three times with TBST, incubated with target protein-specific antibodies overnight at 4 °C, and then incubated with a secondary antibody at ambient temperature for 1 h. Specific protein bands were visualized using chemiluminescence (Peiqing JS-1075 Mini automatic chemiluminescence image analysis system; Peiqing Science and Technology Co., Ltd., Shanghai, China). GAPDH was used as a reference protein.

### 4.9. Statistical Analysis

Data are presented as the mean ± SD. Statistical analysis was conducted using SPSS software version 17.0 (IBM, Inc., Armonk, NY, USA). Differences were analyzed using one-way ANOVA, followed by Tukey’s multiple comparison test. A *p* < 0.05 was considered to represent statistical significance.

## 5. Conclusions

In conclusion, NAOs may relieve the symptoms of T2DM by improving the growth of animals, reducing the fasting blood glucose levels, enhancing glucose tolerances, ameliorating liver damage, decreasing pancreatic β-cell loss, and elevating the antioxidant abilities. The anti-diabetic effects of NAOs can be contributed to their abilities to activate the MAPK, Nrf2, and PPARγ pathways in T2DM mice. NAOs may be useful in the treatment of T2DM due to their capacity to retard some complicated syndromes that occur with T2DM.

## Figures and Tables

**Figure 1 marinedrugs-17-00541-f001:**
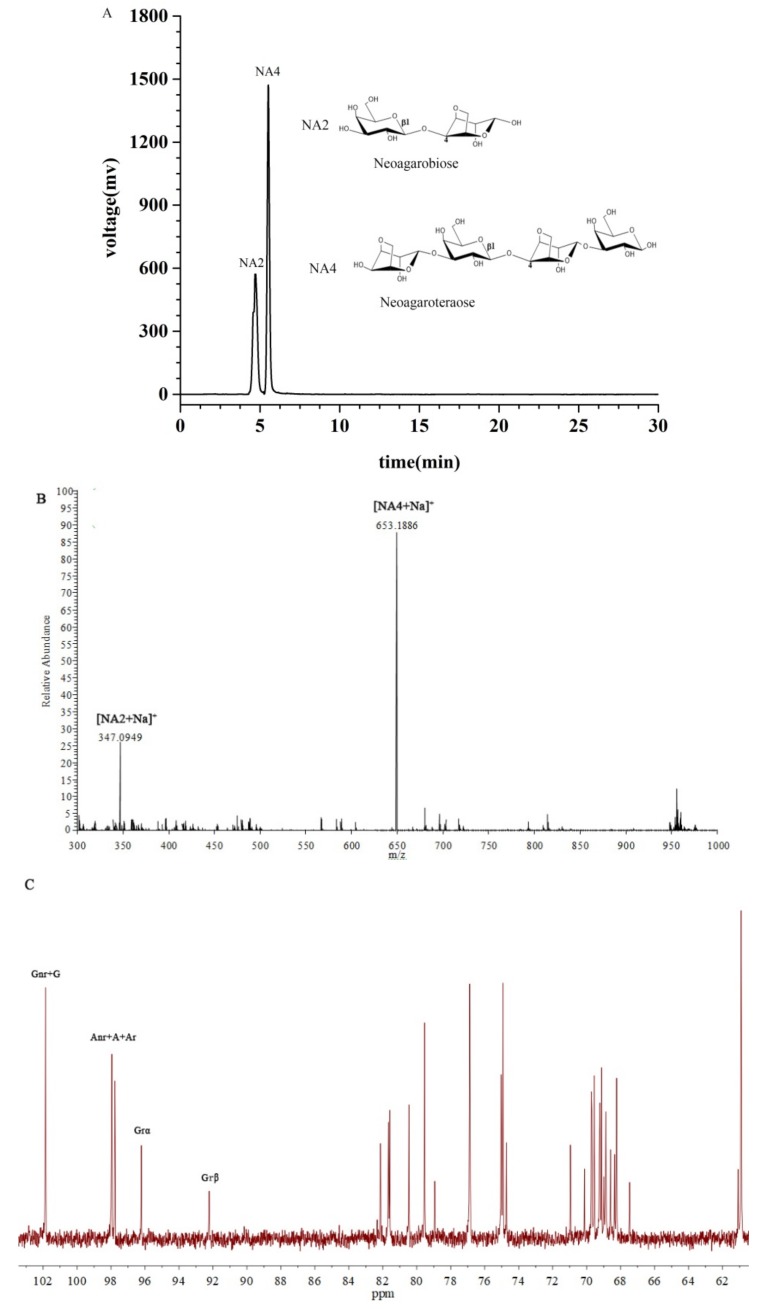
(**A**) HPLC results, (**B**) ESIMS spectra results, and (**C**) ^13^C NMR spectra results of the neoagaro-oligosaccharides (NAOs).

**Figure 2 marinedrugs-17-00541-f002:**
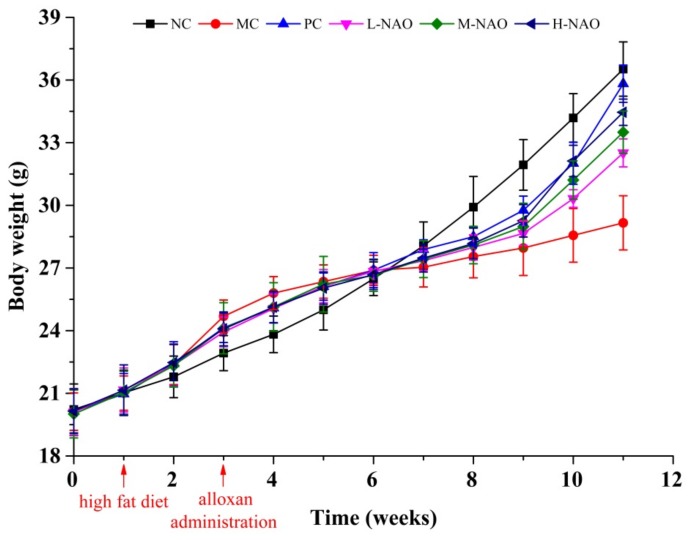
Body weight of KM mice. Normal control group (NC), model control group (MC), positive control group (PC), low-dose NAOs group (L-NAO), medium-dose NAOs group (M-NAO), and high-dose NAOs group (H-NAO).

**Figure 3 marinedrugs-17-00541-f003:**
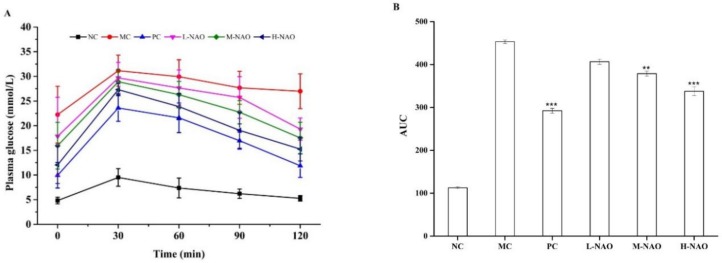
Plasma glucose levels of T2DM mice after the oral glucose tolerance test (OGTT) (**A**) and the area under the curve (AUC) (**B**). ∗∗ *p* < 0.01 and ∗∗∗ *p* < 0.001 compared with the model control group (MC); normal control group (NC), positive control group (PC), low-dose NAOs group (L-NAO), medium-dose NAOs group (M-NAO), and high-dose NAOs group (H-NAO).

**Figure 4 marinedrugs-17-00541-f004:**
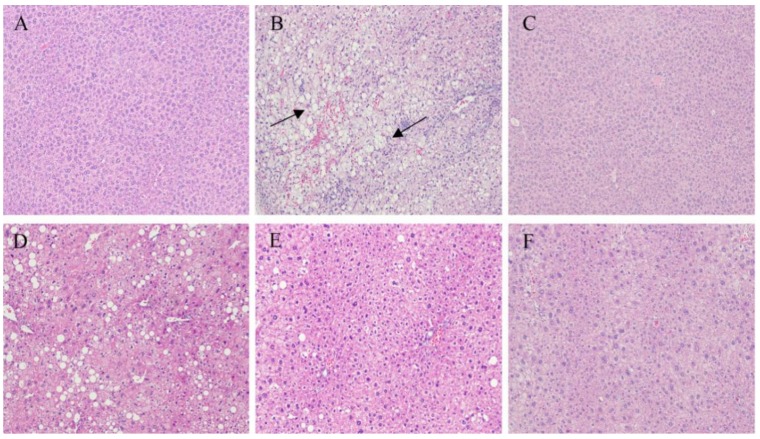
Histology properties of the mice liver sections obtained from (**A**) the normal control group (NC), (**B**) model control group (MC), (**C**) positive control group (PC), (**D**) low-dose NAOs group (L-NAO), (**E**) medium-dose NAOs group (M-NAO), and (**F**) high-dose NAOs group (H-NAO). Mice were fed normal saline, HFD, metformin, or 100 mg/kg, 200 mg/kg, or 400 mg/kg NAOs, respectively. Magnification: 100×. The arrows indicate fatty hepatocytes.

**Figure 5 marinedrugs-17-00541-f005:**
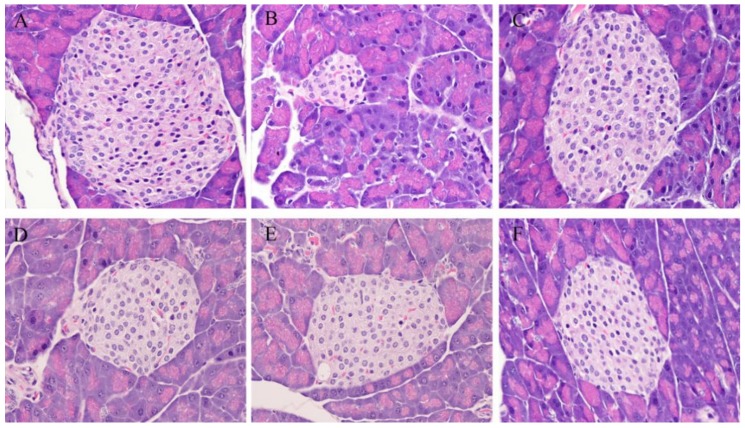
Histological images of pancreatic tissue sections. (**A**) Normal diet. (**B**) Diabetes without treatment. (**C**) Diabetes and treatment with 50 mg/kg metformin. (**D**) Diabetes and treatment with 100 mg/kg NAOs. (**E**) Diabetes and treatment with 200 mg/kg NAOs. (**F**) Diabetes and treatment with 400 mg/kg NAOs. B, C, D, E, and F groups were fed a HFD. Magnification: 200×.

**Figure 6 marinedrugs-17-00541-f006:**
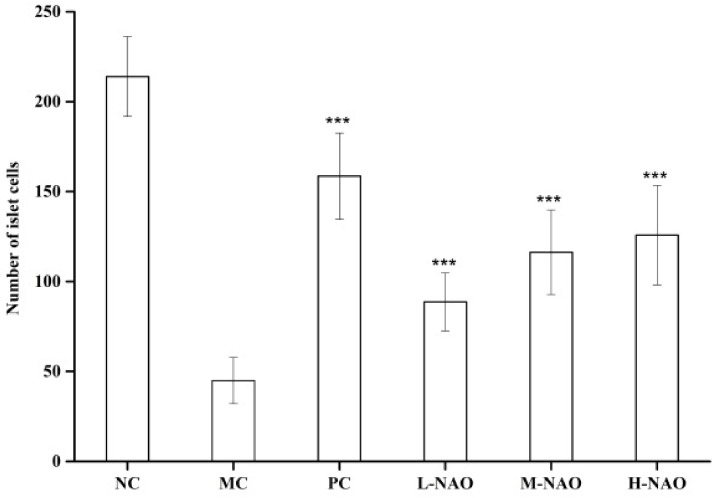
The number of islet cells in pancreatic tissues. Normal control group (NC), model control group (MC), positive control group (PC), low-dose NAOs group (L-NAO), medium-dose NAOs group (M-NAO), and high-dose NAOs group (H-NAO).

**Figure 7 marinedrugs-17-00541-f007:**
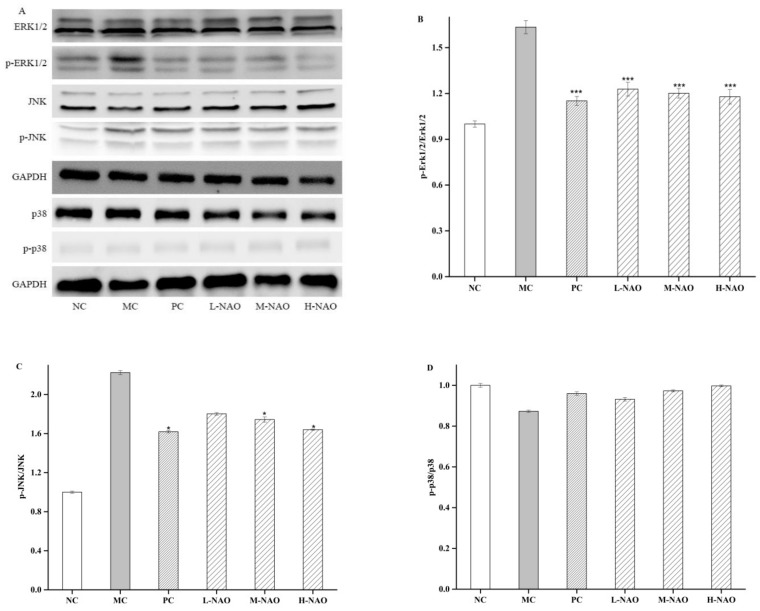
Effects of NAOs on MAPK pathways in mice liver. ∗ *p* < 0.05 and ∗∗ *p* < 0.01 compared with the model control group (MC); normal control group (NC), positive control group (PC), low-dose NAOs group (L-NAO), medium-dose NAOs group (M-NAO), and high-dose NAOs group (H-NAO). (**A**) After pretreated with NAOs, the expression levels of ERK1/2, JNK, p38 and phosphorylated ERK1/2, JNK, p38 were detected by western blot, respectively. Blots were also probed for GAPDH as loading controls. (**B**) Quantification of immunoblot for the ratio of phosphorylated ERK1/2 to ERK1/2. (**C**) Quantification of immunoblot for the ratio of phosphorylated JNK to JNK. (**D**) Quantification of immunoblot for the ratio of phosphorylated p38 to p38.

**Figure 8 marinedrugs-17-00541-f008:**
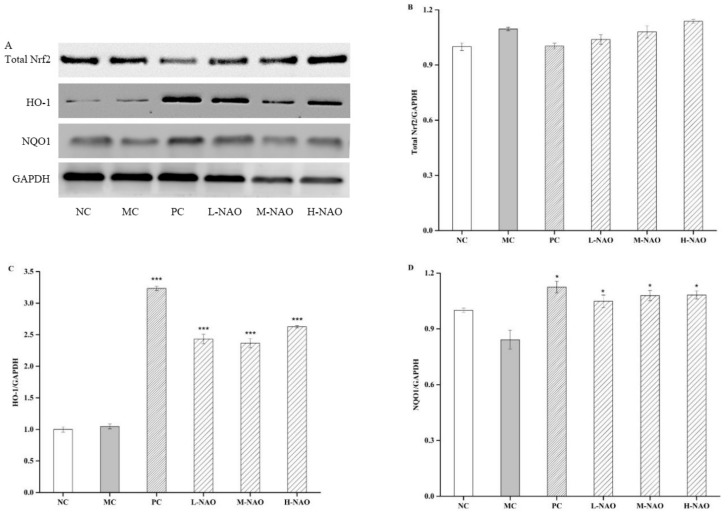
Effects of NAOs on Nrf2, HO-1, and NQO1 levels in mice liver. ∗ *p* < 0.05, ∗∗ *p* < 0.01 and ∗∗∗ *p* < 0.001 compared with the model control group (MC); normal control group (NC), positive control group (PC), low-dose NAOs group (L-NAO), medium-dose NAOs group (M-NAO), and high-dose NAOs group (H-NAO). (**A**) After pretreated with NAOs, the expression levels of Total Nrf2, HO-1 and NQO1 were detected by western blot, respectively. Blot was also probed for GAPDH as loading controls. (**B**) Quantification of immunoblot for the ratio of Total Nrf2 to GAPDH (**C**) Quantification of immunoblot for the ratio of HO-1 to GAPDH. (**D**) Quantification of immunoblot for the ratio of NQO1 to GAPDH.

**Figure 9 marinedrugs-17-00541-f009:**
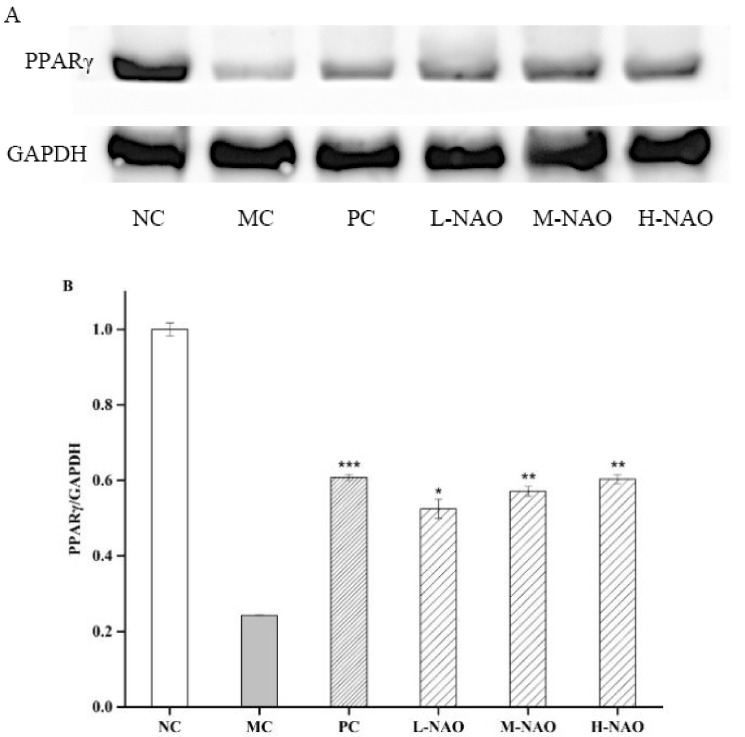
Effects of NAOs on the PPAR γ level in mice liver. ∗ *p* < 0.05, ∗∗ *p* < 0.01, and ∗∗∗ *p* < 0.001 compared with the model control group (MC); normal control group (NC), positive control group (PC), low-dose NAOs group (L-NAO), medium-dose NAOs group (M-NAO), and high-dose NAOs group (H-NAO). (**A**) After pretreated with NAOs, the expression levels of PPAR γ was detected by western blot. Blot was also probed for GAPDH as loading controls. (**B**) Quantification of immunoblot for the ratio of PPAR γ to GAPDH.

**Table 1 marinedrugs-17-00541-t001:** Effect of NAOs on body weight and food intake of type 2 diabetes mellitus (T2DM) mice.

Group			Body Weight (g)		Food Intake (g/day)
	0 Week	1 Week (HFD)	3 Weeks (Alloxan Administration)	11 Weeks	
NC	20.23 ± 1.23	21.03 ± 1.05	22.93 ± 0.84	36.53 ± 1.30	5.16 ± 0.28
MC	20.13 ± 0.90	21.01 ± 0.82	24.69 ± 0.78 ^##^	29.16 ± 1.30 ^##^	4.60 ± 0.29
PC	20.15 ± 1.05	20.98 ± 0.98	24.14 ± 0.70	35.82 ± 0.89 **	4.61 ± 0.27
L-NAO	20.10 ± 1.13	21.16 ± 1.05	23.94 ± 0.71	32.51 ± 0.67 **	4.42 ± 0.24
M-NAO	20.01 ± 1.15	21.04 ± 1.08	24.11 ± 1.23	33.50 ± 1.01 **	4.43 ± 0.22
H-NAO	20.15 ± 1.06	21.15 ± 1.21	24.09 ± 0.81	34.46 ± 0.63 **	4.39 ± 0.17

∗ *p* < 0.05 and ∗∗ *p* < 0.01 compared with the model control group (MC); ^#^
*p* < 0.05 and ^##^
*p* < 0.01 compared with the normal control group (NC); positive control group (PC), low-dose NAOs group (L-NAO), medium-dose NAOs group (M-NAO), and high-dose NAOs group (H-NAO); high-fat diet (HFD).

**Table 2 marinedrugs-17-00541-t002:** Glucose concentration of the KM mice (mmol/L).

Group	0 Week	1 Week (HFD)	3 Weeks (Alloxan Administration)	7 Weeks	11 Weeks
NC	4.44 ± 0.49	4.59 ± 0.59	4.44 ± 0.49	4.10 ± 0.43	4.83 ± 0.69
MC	4.18 ± 0.56	4.53 ± 0.49	22.90 ± 6.16 ^##^	22.96 ± 4.80 ^##^	22.24 ± 5.76 ^##^
PC	4.33 ± 0.52	4.44 ± 0.45	22.60 ± 5.69	17.79 ± 8.63	9.95 ± 2.59 **
L-NAO	4.68 ± 0.44	4.59 ± 0.51	22.63 ± 5.68	17.94 ± 10.62	17.85 ± 7.90
M-NAO	4.33 ± 0.50	4.55 ± 0.49	22.46 ± 5.55	18.14 ± 8.42	15.95 ± 4.72 *
H-NAO	4.33 ± 0.46	4.73 ± 0.32	22.44 ± 5.34	14.89 ±6.46 *	12.06 ± 3.77 **

∗ *p* < 0.05 and ∗∗ *p*
*<* 0.01 compared with the model control group (MC); ^#^
*p*
*<* 0.05 and ^##^
*p*
*<* 0.01 compared with the normal control group (NC); positive control group (PC), low-dose NAOs group (L-NAO), medium-dose NAOs group (M-NAO), and high-dose NAOs group (H-NAO); high-fat diet (HFD).

**Table 3 marinedrugs-17-00541-t003:** Serum lipoproteins in mice.

Group	TC (mmol/L)	TG (mmol/L)	HDL-C (mmol/L)	LDL-C (mmol/L)
NC	2.60 ± 0.24	0.91 ± 0.08	2.76 ± 0.34	1.67 ± 0.20
MC	5.59 ± 0.22 ^##^	2.01 ± 0.06 ^##^	1.91 ± 0.27 ^##^	3.60 ± 0.23 ^##^
PC	2.83 ± 0.07 **	1.21 ± 0.05 **	2.87 ± 0.19 **	1.79 ± 0.22 **
L-NAO	5.39 ± 0.13	1.94 ± 0.09	2.01 ± 0.07	3.52 ± 0.45
M-NAO	5.04 ± 0.14 *	1.87 ± 0.06 *	2.03 ± 0.41	3.08 ± 0.32 *
H-NAO	4.60 ± 0.28 **	1.35 ± 0.05 **	2.33 ± 0.18 *	2.91 ± 0.32 *

∗ *p* < 0.05 and ∗∗ *p* < 0.01 compared with the model control group (MC); ^#^
*p* < 0.05 and ^##^
*p* < 0.01 compared with the normal control group (NC); positive control group (PC), low-dose NAOs group (L-NAO), medium-dose NAOs group (M-NAO), and high-dose NAOs group (H-NAO). TC: total cholesterol; TG: triglycerides; HDL-C; high-density lipoprotein cholesterol; LDL-C: low-density lipoprotein cholesterol.

**Table 4 marinedrugs-17-00541-t004:** Antioxidative activities of NAOs in mice liver.

Groups	MDA (nmol/g Protein)	GSH-Px (U/mg Protein)	SOD (U/mg Protein)
NC	71.37 ± 0.96	13.02 ± 0.47	82.48 ± 1.85
MC	92.01 ± 5.68 ^##^	9.65 ± 0.26 ^##^	52.37 ± 2.85 ^##^
PC	75.54 ± 3.82 **	12.34 ± 0.43 **	75.84 ± 1.41 **
L-NAO	89.83 ± 3.82	10.18 ± 0.42 *	55.02 ± 2.53
M-NAO	85.41 ± 4.67 *	10.82 ± 0.31 **	60.62 ± 1.34 **
H-NAO	79.61 ± 2.32 **	11.41 ± 0.58 **	65.54 ± 2.88 **

∗ *p* < 0.05 and ∗∗ *p* < 0.01 compared with the model control group (MC); ^#^
*p* < 0.05 and ^##^
*p* < 0.01 compared with the normal control group (NC); positive control group (PC), low-dose NAOs group (L-NAO), medium-dose NAOs group (M-NAO), and high-dose NAOs group (H-NAO).

**Table 5 marinedrugs-17-00541-t005:** Composition of high-fat diet.

Constituents	gm%	kcal%
Protein	24	20
Carbohydrate	41	35
Fat	24	45
Ingredients	gm	kcal
Casein, 30 Mesh	200	800
L-Cystine	3	12
Corn Starch	72.8	271
Maltodextrin 10	100	400
Sucrose	172.8	691
Cellulose, BW200	50	0
Soybean Oil	25	225
Lard	177.5	1598
Mineral Mix S10026	10	0
DiCalcium Phosphate	13	0
Calcium Carbonate	5.5	0
Potassium Citrate, 1 H_2_O	16.5	0
Vitamin Mix V10001	10	40
Choline Bitartrate	2	0
FD&C Red Dye #40	0.05	0

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
