# Peer review of "The Potential of Neoagaro-Oligosaccharides as a Treatment of Type II Diabetes in Mice"

_marinedrugs, 2019, doi:10.3390/md17100541_

Round 1

Reviewer 1 Report

Overall this is a good study, exploring NAO’s as potential treatment of T2DM. Results from different experiments are supportive of possible role of NAO’s in ameliorating T2DM symptoms. However, the data for MAPK pathway intermediates phosphorylation should be reanalyzed and results should be interpreted accordingly. Levels of total phosphorylated protein should be calculated over total respective protein levels, not GAPDH. Looking at the blots it seems that trends will differ for the three proteins tested. Authors should look for p-protein/total respective protein levels for all phosphorylate proteins (p-ERK1/2/ERK1/2, p-JNK/JNK, p-p-38/p-38) protein, not GAPDH.

Author Response

Response: Thanks a lot for your advice. It is right and helpful. The results have been expressed as the ratio of p-protein/respective protein according to the reviewer. In addition, we have reanalyzed the MAPK pathway and the discussion has been interpreted accordingly. HFD and alloxan resulted in higher phosphorylation of ERK1/2 and JNK as compared with the NC group. But 8 weeks of NAOs treatment significantly ameliorated the abnormal phosphorylation of ERK1/2 and JNK, with little change of p-p38MAPK, which statistical analysis was conducted. It protects mice’s liver from the oxidative damage induced by high blood glucose.

Reviewer 2 Report

I think that the oligosaccharide mixture used in the experiments was characterised insufficiently. I recommend to add more analytical data (elemental analysis, NMR, MS etc.) clarifying purity, composition and structure of these oligosaccharides. Also, there are several minor points for revision:

Page 8, lines 201-207: this part should be delete or moved to the Introduction. Page 8, lines 208-209: references from literature should be added. Page 10, Materials and animals: the origin of agar used for preparation of oligosaccharides should be specified (source, purity, composition, producer etc.).

Author Response

Response: Thank you for your comments. According to the reviewer, we added the results of high resolution ESI-MS and 13C-NMR spectra to the revised manuscript (Figure 1B and Figure 1C). The found m/s results match the calculated ones very well, confirming the structures of NAOs. The distinctive chemical shifts (92.6 ppm and 96.6 ppm) were easily found in the 13C-NMR spectra, indicating the existence of the α- and β-anomeric ends of the NAOs molecules. That was, the results shown in the revised Figure 1 might benefit to understanding the purity, composition and structure of these oligosaccharides.

Page 8, lines 201-207 has been deleted accordingly.

Page 8, lines 208-209. According to the reviewer, some references have been added to the revision as followed:

Hong, S. J.; Lee, J. H.; Kim, E. J.; Yang, H. J.; Park, J. S.; Hong, S. K., Anti-Obesity and Anti-Diabetic Effect of Neoagarooligosaccharides on High-Fat Diet-Induced Obesity in Mice. Marine drugs 2017, 15, (4). Yang, J. H.; Cho, S. S.; Kim, K. M.; Kim, J. Y.; Kim, E. J.; Park, E. Y.; Lee, J. H.; Ki, S. H., Neoagarooligosaccharides enhance the level and efficiency of LDL receptor and improve cholesterol homeostasis. Journal of Functional Foods 2017, 38, 529-539.

For page 10. The origin, purity, composition and producer of the agar used for preparation of oligosaccharides have been presented in the revised manuscript.

Round 2

Reviewer 2 Report

The revised article is acceptable for publication.